# Emergence of artemisinin-resistant *Plasmodium falciparum* with *kelch13* C580Y mutations on the island of New Guinea

**Olivo Miotto**[1,2,3,4]ᴼ*, **Makoto Sekihara**[5]ᴼ, **Shin-Ichiro Tachibana**[5], **Masato Yamauchi**[5], **Richard D. Pearson**[1,3], **Roberto Amato**[3], **Sonia Gonçalves**[3], **Somya Mehra**[6], **Rintis Noviyanti**[7], **Jutta Marfurt**[8], **Sarah Auburn**[2,4,8], **Ric N. Price**[2,4,8], **Ivo Mueller**[6], **Mie Ikeda**[5], **Toshiyuki Mori**[5], **Makoto Hirai**[5], **Livingstone Tavul**[9], **Manuel W. Hetzel**[10,11], **Moses Laman**[9], **Alyssa E. Barry**[6,12,13,14], **Pascal Ringwald**[15], **Jun Ohashi**[16], **Francis Hombhanje**[17], **Dominic P. Kwiatkowski**[1,3], **Toshihiro Mita**[5]*

**1** MRC Centre for Genomics and Global Health, Big Data Institute, University of Oxford, Oxford, United Kingdom, **2** Mahidol-Oxford Tropical Medicine Research Unit, Mahidol University, Bangkok, Thailand, **3** Wellcome Sanger Institute, Hinxton, United Kingdom, **4** Centre for Tropical Medicine and Global Health, Nuffield Department of Clinical Medicine, University of Oxford, Oxford, United Kingdom, **5** Department of Tropical Medicine and Parasitology, Juntendo University Faculty of Medicine, Tokyo, Japan, **6** Walter and Eliza Hall Institute of Medical Research, Melbourne, Australia, **7** Eijkman Institute for Molecular Biology, Jakarta, Indonesia, **8** Global and Tropical Health Division, Menzies School of Health Research and Charles Darwin University, Darwin, Australia, **9** Papua New Guinea Institute of Medical Research, Madang, Papua New Guinea, **10** Swiss Tropical and Public Health Institute, Basel, Switzerland, **11** University of Basel, Basel, Switzerland, **12** University of Melbourne, Melbourne, Australia, **13** Institute for Mental and Physical Health and Clinical Translation (IMPACT), Deakin University, Geelong, Australia, **14** Burnet Institute, Melbourne, Australia, **15** World Health Organization, Geneva, Switzerland, **16** Department of Biological Sciences, Graduate School of Science, University of Tokyo, Tokyo, Japan, **17** Centre for Health Research & Diagnostics, Divine Word University, Madang, Papua New Guinea

ᴼ These authors contributed equally to this work.
* olivo@tropmedres.ac (OM); tmita@juntendo.ac.jp (TM)

**Data Availability Statement:** Sequence data are deposited in the European Nucleotide Archive (ENA, https://www.ebi.ac.uk/ena), and publicly available. Identifiers are provided in S7 Table.

## Abstract

The rapid and aggressive spread of artemisinin-resistant *Plasmodium falciparum* carrying the C580Y mutation in the *kelch13* gene is a growing threat to malaria elimination in Southeast Asia, but there is no evidence of their spread to other regions. We conducted cross-sectional surveys in 2016 and 2017 at two clinics in Wewak, Papua New Guinea (PNG) where we identified three infections caused by C580Y mutants among 239 genotyped clinical samples. One of these mutants exhibited the highest survival rate (6.8%) among all parasites surveyed in ring-stage survival assays (RSA) for artemisinin. Analyses of *kelch13* flanking regions, and comparisons of deep sequencing data from 389 clinical samples from PNG, Indonesian Papua and Western Cambodia, suggested an independent origin of the Wewak C580Y mutation, showing that the mutants possess several distinctive genetic features. Identity by descent (IBD) showed that multiple portions of the mutants' genomes share a common origin with parasites found in Indonesian Papua, comprising several mutations within genes previously associated with drug resistance, such as *mdr1*, *ferredoxin*, *atg18* and *pnp*. These findings suggest that a *P. falciparum* lineage circulating on the island of New Guinea has gradually acquired a complex ensemble of variants, including *kelch13* C580Y, which have affected the parasites' drug sensitivity. This worrying development

**Funding:** This study was supported by: - Grants-in-aid for scientific research from the Japan Society for the Promotion of Science and Foundation of Strategic Research Projects in Private Universities (TM; https://www.jsps.go.jp/english/e-grants/; 26460515, 26305015, 17H04074, 18KK0231) - The Ministry of Education, Culture, Sports, Science, and Technology of Japan (TM; https://www.mext.go.jp/en/; S0991013) - Japan Agency for Medical Research and Development (TM; https://www.amed.go.jp/en/; JP15km0908001) - The Global Health Innovative Technology Fund (TM; https://www.ghitfund.org/; G2015-210) - The Global Fund to Fight AIDS, Tuberculosis and Malaria (MWH, AEB; https://www.theglobalfund.org/en/) - The National Health and Medical Research Council of Australia (AEB; https://www.nhmrc.gov.au/; GNT1027108, APP 1131932) - The Australian Centre for Research Excellence on Malaria Elimination (SA; https://www.acreme.org.au/; APP 1134989) - The Australian Government Department of Foreign Affairs and Trade through the Tropical Disease Research Regional Collaboration Initiative (AEB; https://indopacifichealthsecurity.dfat.gov.au/TDRRCI) - The Victorian State Government Operational Infrastructure Support (AEB; https://www2.health.vic.gov.au/about/clinical-trials-and-research/operational-infrastructure-support) - The Wellcome Trust (DPK; https://wellcome.ac.uk/; 206194, 090770/Z/09/Z) - The Bill & Melinda Gates Foundation (OM, DPK; https://www.gatesfoundation.org/; OPP11188166) - The Medical Research Council of the UK (DPK; https://mrc.ukri.org/; G0600718) The funders had no role in study design, data collection and analysis, decision to publish, or preparation of the manuscript.

**Competing interests:** The authors have declared that no competing interests exist.

reinforces the need for increased surveillance of the evolving parasite populations on the island, to contain the spread of resistance.

## Author summary

Artemisinin is the most widely used drug against *Plasmodium falciparum* malaria. In southeast Asia, parasites have evolved genetic changes making them resistant to artemisinin. The elimination of resistant strains is a global priority, since their global spread could result in massive loss of lives. In Papua New Guinea, we found three patients infected with parasites carrying the most widespread resistant variant in southeast Asia, and they were confirmed to be artemisinin resistant. We established that the mutations were not imported from southeast Asia, and found other drug resistance variants in their genetic background, including some shared with parasites in Indonesia. This indicates that artemisinin resistance has emerged in New Guinea separately from southeast Asia, not by a chance event, but by a gradual process of evolution which may still be ongoing undetected on the island. These resistant strains could undermine malaria local control efforts, and constitute a global threat.

## Introduction

The global adoption of artemisinin-based combination treatments (ACTs), consisting of an artemisinin derivative and a partner drug, has played a key role in the worldwide reduction of malaria cases and deaths [1]. In the last decade, however, *Plasmodium falciparum* parasites with decreased sensitivity to artemisinin have emerged at multiple locations in the Greater Mekong Sub-Region (GMS) [2–4]. These resistant parasites exhibit a reduced clearing rate during treatment [2,3,5], such that treatment efficacy relies more heavily on the partner drug. As a result, resistance to the partner drug may emerge, as has recently been the case with the partner drug piperaquine, leading to ACT treatment failures [4,6].

In the GMS, slow artemisinin clearance rates have been associated with genetic mutations in two domains of the parasite's *kelch13* gene [7]. Numerous independently arising *kelch13* mutations have been found in GMS resistant parasites [3,8–10], and several of these mutations have been validated in relation with clinical and *in vitro* artemisinin resistance [11]. Increasingly, the genotyping of *kelch13* is being used for the surveillance of artemisinin resistance [3,8–10]. After an initial period in which many *kelch13* resistant alleles circulated concurrently, diversity has declined dramatically in the eastern part of the GMS (comprising Cambodia, northeast Thailand, southern Laos and Vietnam) where the C580Y allele has become the most prevalent, replacing other circulating mutations [12]. This allele is also associated with reduced clearing rate western Thailand [5] where it has emerged independently [13]. The C580Y allele has also been detected in parasites outside the GMS, in Africa [8] and in Guyana, South America [14]; however, in these instances the artemisinin clearance rates were not known.

While the GMS has been at the epicentre of resistance to antimalarials multiple times over the last sixty years, resistance has also repeatedly arisen independently on the island of New Guinea, producing alternative resistant genetic variants [15,16]; This makes New Guinea a likely candidate for the emergence of novel forms of artemisinin resistance, and underlines the importance of monitoring local strains. The island of New Guinea is divided between the countries of Papua New Guinea (also known as "PNG") to the east, and Indonesia to the west

(referred to as "Indonesian Papua" here). PNG has used artemether-lumefantrine as first-line treatment for malaria since late 2011[17], and delayed parasite clearance has not been reported to date [18]. We previously reported the absence of *kelch13* mutations in PNG samples obtained in 2002–2003 in Wewak Town, East Sepik [9], consistent with results from a later broader survey [10]; however, studies of genetic variants across the island have only been conducted at a limited number of sites. A report of an Australian citizen infected with a C580Y mutant after returning from PNG raised the prospect that artemisinin resistance may have emerged undetected [19]. To investigate this possibility, we conducted a cross-sectional study in 2016–2017 in Wewak. Here, we analyze clinical and genetic data from 257 cases surveyed in this study, to evaluate the potential emergence of artemisinin resistance. We assessed sensitivity to artemisinin using the ring-stage survival assay (RSA), a sensitive ex-vivo method [20], and used next-generation sequencing data to reconstruct the epidemiology of *kelch13* mutants found.

## Results

### Identification and characterization of *kelch13* C580Y mutants

This study enrolling 257 patients with uncomplicated malaria in 2016 and 2017 at two clinics in Wewak Town, East Sepik in PNG (Fig 1). Eighteen patients (7.0%) had taken antimalarial drugs in the previous two weeks, and were excluded from further analysis (Table 1). We were able to genotype the *kelch13* gene in 239 parasitized blood samples (S1 Table). The C580Y allele was found in 2017 in three samples (2.3%), labelled PNG-C580Y-1 to PNG-C580Y-3 (S2 Table). The patients carrying the mutant parasites were aged 7, 16 and 25 years, and presented with parasite densities of 0.35, 0.07 and 0.44%, respectively. They lived within 2 km of each other and had no history of foreign travel (S1 Fig).

We used RSA [20] to measure artemisinin susceptibility in samples with parasitemia >0.1% (n = 117), including two C580Y mutants. The assays yielded interpretable results for 57

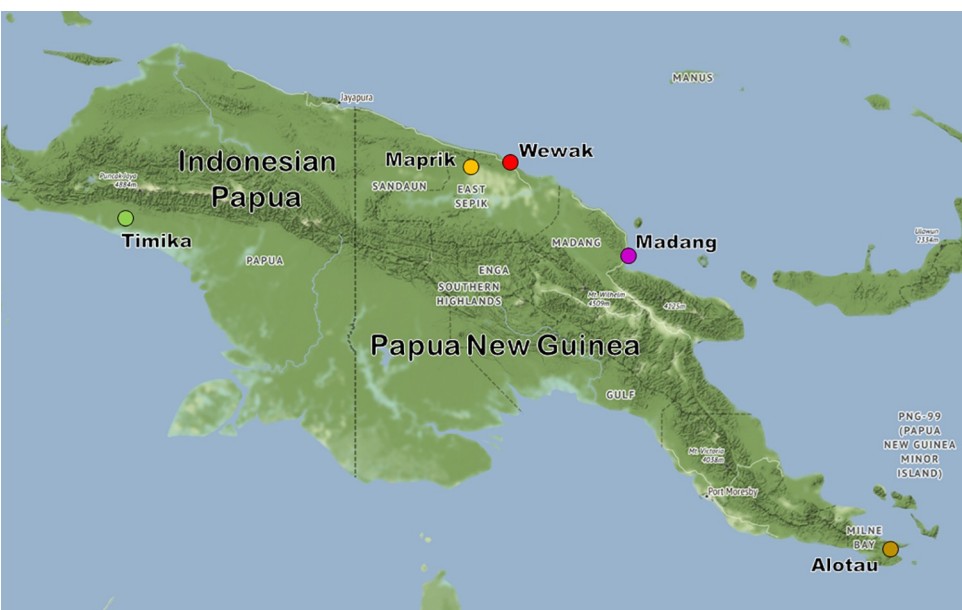

**Fig 1. Geographical location of New Guinea sites.** Sites that contributed samples in both countries that form part of the island of New Guinea (Indonesia to the west and PNG to the east) are shown by coloured circular markers. Map tiles by Stamen Design (stamen.com), under CC-BY 3.0. Data ©OpenStreetMap contributors (openstreetmap.org).

**Table 1. Characteristics of enrolled patients.**

| Characteristics | | 2016 | 2017 |
|---|---|---|---|
| Patients (n) | | 123 | 134 |
| Gender (n)[1] | | | |
| | Male | 53 | 57 |
| | Female | 68 | 76 |
| Age (years) | | | |
| | Median (IQR)[2] | 20 (13, 33.5) | 18 (11, 24) |
| Pretreatment | | | |
| | Artemether | 3 (2.5%) | 4 (2.8%) |
| | Artemether/Lumefantrine | 0 | 2 (1.3%) |
| | Artemether/Lumefantrine+Primaquine | 0 | 1 (0.7%) |
| | Chloroquine | 2 (1.7%) | 5 (3.4%) |
| | Primaquine | 1 (0.8%) | 0 |
| Parasitemia | | | |
| | Geometric mean (Range) | 0.11% (0.0007%-9.47%) | 0.28% (0.004%-5.35%) |
| | Median (IQR)[2] | 0.14% (0.02%-0.52%) | 0.33% (0.1%-0.88%) |

[1] Unknown in three individuals

[2] IQR: Interquartile range

parasites (S3 Table), including PNG-C580Y-1. Undetermined results were mainly due to insufficient parasite growth (n = 52) or low-quality blood smears (n = 8). Survival rate could not be determined for PNG-C580Y-3. Five samples (8.8%) exhibited a survival rate >1% (range: 1.6–6.8%) at 72 hours following dihydroartemisinin exposure, a threshold that correlates with parasite clearance half-life >5 hours in the GMS [20]. The PNG-C580Y-1 mutant produced the highest survival rate (6.8%), comparable to rates measured in Cambodian C580Y parasites [7], suggesting that Wewak C580Y mutants possess levels of ring-stage artemisinin resistance similar to those in the GMS (S2 Fig).

## Assessment of *kelch13* mutants' genetic background

To investigate whether the Wewak C580Y mutants were imported from Southeast Asia, we first tested for the genetic background underpinning *kelch13* mutations in the GMS [21]. Of six markers tested, only one (*ferredoxin* D193Y) was found in Wewak C580Y mutants, compared to four or five in GMS artemisinin-resistant clones (S4 Table). The *ferredoxin* variant was detected in all three Wewak C580Y mutants, but was only found in one in ten randomly selected Wewak parasites with wild-type *kelch13*. All 2017 samples were tested for amplifications of *plasmepsin2/3* and *pfmdr1*, which are associated with resistance to the partner drugs piperaquine and mefloquine. All carried a single copy of *plasmepsin2/3*, and only a single parasite (*kelch13* wild-type) had an amplified *pfmdr1* (S1 Table).

## Genome-level relationship of *kelch13* mutants with other populations

To study in greater detail the provenance of the C580Y mutants, we compared whole genome sequencing (WGS) data from 389 high-quality samples with low within-sample diversity, including 73 from the present study, 83 from other PNG locations, 88 from Timika in Indonesian Papua, and 145 from western Cambodia (Table 2 and Fig 1). All three C580Y mutants were included in this analysis, and no other parasite from Indonesia or PNG carried *kelch13* mutations.

**Table 2. Samples used in whole-genome comparative analyses.**

| Country | Region | Location | Sample Count |
|---|---|---|---|
| PNG | East Sepik | Wewak | 73 |
| | | Maprik | 39 |
| | Madang | Madang | 20 |
| | Milne Bay | Alotau | 24 |
| Indonesia | Papua | Timika | 88 |
| Cambodia | Western Cambodia | Pailin | 70 |
| | | Pursat | 75 |
| | | **Total** | **389** |

To cluster individuals by genetic similarity, we conducted a principal coordinates analysis (PCoA) based on genome-wide pairwise genetic distances. This showed a clear separation between Cambodian and New Guinean parasites, with no samples bridging the gap between these two populations (S3 Fig). We also conducted a PCoA without Cambodian parasites (Fig 2A), in which the first component (PC1) clearly separated Papua Indonesia from PNG, showing that parasite populations at the PNG sites are more similar to each other than they are to Indonesian parasites. However, a number of individuals from both countries, including the Wewak C580Y parasites, occupy an intermediate position along PC1, consistent with admixture between the two populations. Furthermore, we observe two Timika parasites clustering with PNG parasites, suggesting recent importation from PNG into Timika.

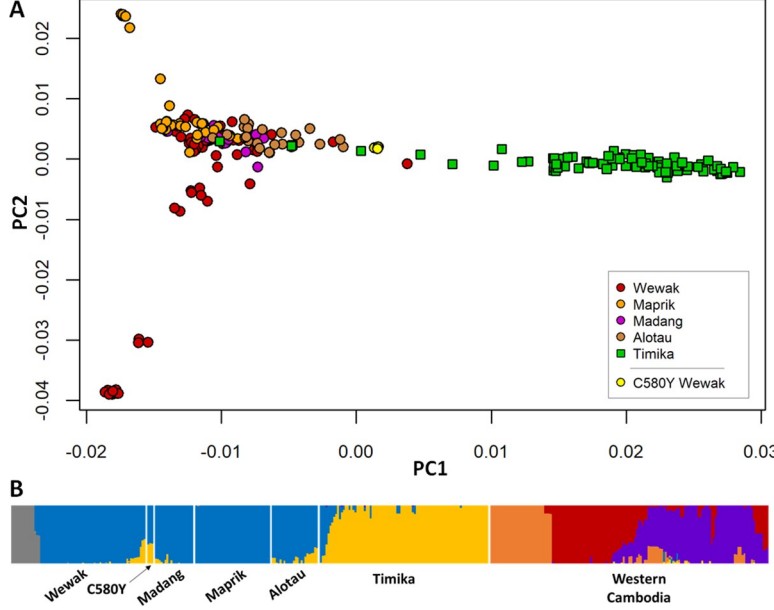

**Fig 2. Population Structure in the New Guinea.** (A) Plot of the first two components (PC2 vs PC1) of principal coordinates analysis (PCoA) using genetic distances computed from whole-genome sequence data from all samples in New Guinea. We note a marked separation between parasites from Papua Indonesia (green squares) and PNG (circles), with a minority of samples from both regions at an intermediate position, suggesting admixture. (B) Population structure at all sites, estimated by fastSTRUCTURE, based on the hypothesis of K = 6 ancestral populations; each ancestral populations is indicated by a different colour, and each vertical bar represents a single sample, coloured according to the estimated ancestry proportions. The plot suggests no recent common ancestry between New Guinea parasites and those in Cambodia. Furthermore, it shows strong differentiations between parasites in Papua Indonesia and PNG (consistent with PCoA results above), with a minority of samples from both sides (including the Wewak C580Y mutants) exhibiting significant levels of admixture. This analysis also identified a subpopulation of Wewak parasites (gray) which correspond to the cluster of samples in the lower left-hand corner of the PCoA plot.

Further evidence of admixture was provided by population structure analysis performed by fastSTRUCTURE (S4 Fig). This analysis indicated that the underlying population structure was best explained by K = 6 putative ancestral populations (Fig 2B). Applying this model, we observe that New Guinea parasites mostly derive from two major ancestral populations, one in Papua Indonesia and one in PNG, with the exception of a small subpopulation in Wewak (gray bars). However, substantially admixture in several samples in both countries (e.g. about 16% of samples from both PNG and Papua Indonesia shared >10% ancestry with the dominant group of the other region) suggests frequent exchanges between the two populations. The Wewak C580Y mutants were found to be amongst the most admixed in PNG, all three individuals estimated to have inherited 35–36% of their genome from the Indonesian ancestral population. Like all New Guinea samples, Wewak C580Y mutants carry no contribution from GMS ancestral populations, ruling out the hypothesis that they are recent imports from that region.

## Origins of the Wewak *kelch13* C580Y haplotype

Although the mutants were not recently imported from the GMS, it is still possible that a C580Y mutation from the GMS could have been acquired by New Guinea parasites through recombination, analogous to the *crt* haplotype of Asian origin that circulates in chloroquine-resistant African *P. falciparum* [22]. To investigate this possibility, we tested 12 microsatellite loci and six SNPs in regions flanking *kelch13* [14,23] in the 2017 parasites from Wewak, and eight other isolates for comparison (S5 Table), producing 64 unique haplotypes in 96 parasites. Upstream from *kelch13* (left flank), we found strong similarities between the haplotype sequence of the Wewak C580Y mutants, and those of both Cambodian resistant parasites (e.g. MRA-1236) and PNG wild type haplotypes (e.g. H5 and H13, found in 7 samples). Downstream from *kelch13*, however, the C580Y mutants bear greater similarity to the sequences of PNG haplotypes (e.g. H46 and H48) than to those in Cambodia. Overall, the H5 and H13 haplotypes from PNG were the most similar to those of the mutants, suggesting an independent local emergence rather than a GMS origin.

Similar results were obtained from flanking haplotypes constructed by concatenating WGS genotypes at SNPs with minor allele frequency ≥ 0.01 in a region of 300kbp centred at *kelch13* (S5 Fig). Downstream, the C580Y mutant flanking haplotype is similar to those circulating in Cambodia, but also to those in *kelch13* wild-type parasites in New Guinea, particularly in Timika. Upstream, considerably longer matches (~60kbp) are found in Indonesian and PNG samples than in Cambodian ones. Taken together, the above results provide no support for a GMS origin of the PNG *kelch13* C580Y haplotype. Since the mutant genomes show low similarity to GMS parasites, it is reasonable to conclude that the mutation most likely emerged independently in New Guinea.

## Ancestry reconstruction of the PNG *kelch13* C580Y genomes

The level of admixture in the C580Y mutants suggest substantial genetic contribution from the Indonesian population. One possibility is that the Wewak mutants originate from a geographically intermediate location, and are the result of long-term continuous recombination between Indonesian and PNG populations. A different hypothesis is that the mutants have acquired through recombination, and retained, portions of genomes circulating in Indonesian Papua which provide some selective advantage. The first scenario would result in a random distribution of shared alleles, while the second would likely produce a limited number of long shared haplotypes.

To evaluate these hypotheses, we performed a genome-wide estimation of *identity by descent* (IBD) between each pair of samples. IBD models recombination processes, identifying

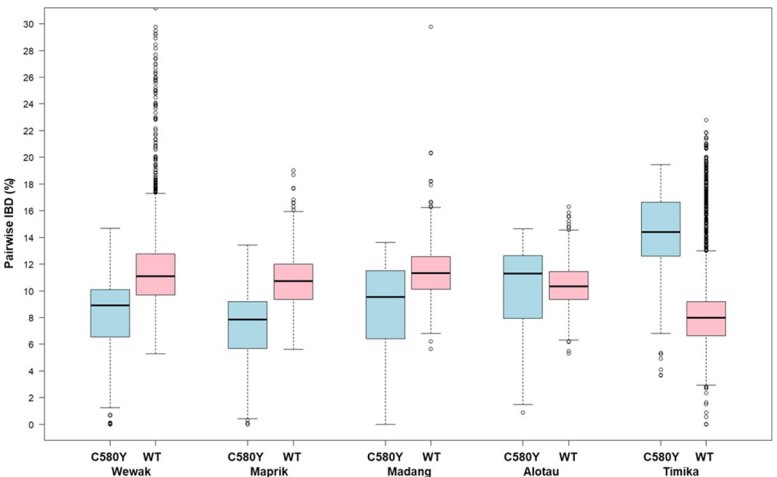

**Fig 3. IBD between parasites from Wewak and those from other New Guinea sites.** This plot shows the distributions of the level of IBD (i.e. the percentage of the genome that is predicted to be in IBD) between parasites samples in Wewak and those sampled at other sites on the island of New Guinea. For each of the five sites, two boxplots show the distribution of pairwise IBD levels for parasites from that site: either against the Wewak C580Y mutants (blue, labelled "C580Y"), or against the Wewak *kelch13* wild-type parasites (pink, "WT"). For the Wewak site (first two boxplots) we compare the two groups against Wewak WT samples only; hence, the blue boxplot shows IBD levels between Wewak C580Y mutants and Wewak WT parasites, while the pink boxplot shows IBD levels among Wewak WT parasites only.

genome regions that are likely to be identical because of common ancestry. For each of the New Guinea sites, we estimated IBD proportion between local parasites and the Wewak C580Y mutants; and between local parasites and the Wewak parasites carrying *kelch13* wild-type (Fig 3). The results indicate that the mutants possess significantly higher IBD with Timika parasites than the Wewak wild-type parasites (medians IBD proportion: 14.4% vs 8.0%, $p<10^{-117}$ by Mann-Whitney test), and significantly less IBD with parasites from most PNG sites (Wewak medians: 8.9% vs 11%, $p<10^{-48}$; Madang medians: 9.5% vs 11.3%, $p<10^{-7}$; Maprik medians: 7.8% vs 10.7%, $p<10^{-37}$). An IBD network based on genome-wide data shows that parasites from Wewak, Timika and Cambodia form three large haplotype-sharing clusters, while the Wewak mutants constitute a small separate group, confirming their genome composition is distinct from these populations (Fig 4A). Also, an IBD network based on the regions flanking the *kelch13* gene shows that the haplotypes accompanying the C580Y mutations are distinct from those circulating elsewhere- including those in Cambodia and Papua Indonesia (Fig 4B).

The observation that Wewak C580Y mutants share a higher proportion of IBD with Timika than with local wild-type parasites might appear to contradict the finding that the proportion of PNG ancestry in these mutants is greater than that of Indonesia, as was determined by fastSTRUCTURE. However, these results could be reconciled if most of the mutants' genomes was produced by long-term mixing of PNG parasites, resulting in a random distribution of alleles common in PNG, and is interspersed with a smaller proportion of segments recently acquired from Indonesian populations, which are identified as IBD. To test this hypothesis, we selected SNPs that are highly differentiated between the Wewak and Timika populations ($F_{ST} \geq 0.3$), and labelled them according to which of the two populations has the allele present in the C580Y mutants as its dominant allele. Of the 467 selected SNPs, 57.6% (n = 269) carried the Wewak allele, consistent with a predominantly PNG-like genetic background. When mapped across the genome, the majority of Timika-like alleles are not randomly distributed, but clustered in regions of IBD between the C580Y mutants and the Timika populations (Fig 5). For example, 63% of variants where the mutants carry the Timika-like allele are in regions

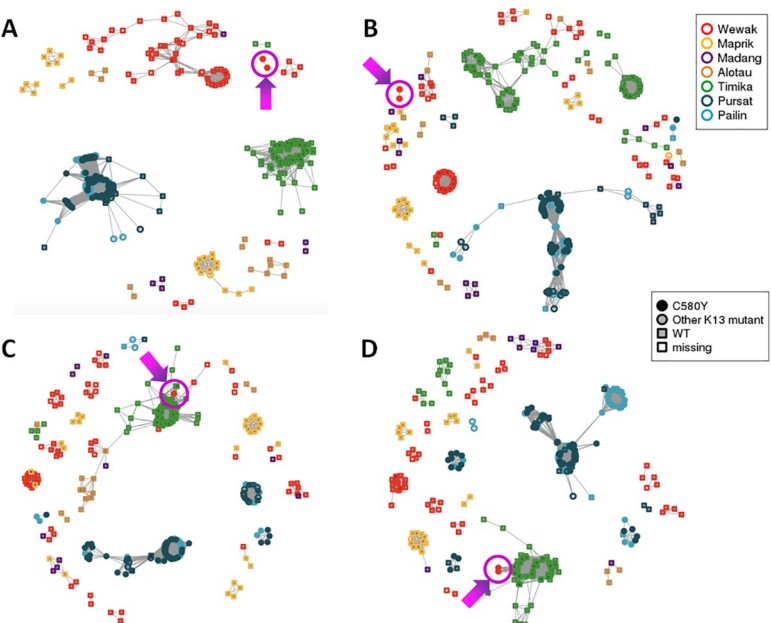

**Fig 4. IDB-based clustering.** These isoRelate network plots show IBD relationships between all samples analyzed for different genomic regions. Wewak C580Y mutants are shown by a magenta circle, pointed to by an arrow; note that only two mutants were included in this analysis, due to high degree of genotype missingness in the third mutant. (A) Whole-genome IBD shows that, although most parasites cluster strongly by geography, the C580Y mutants form a separate group, probably reflecting their admixed structure. (B) IBD in chromosome 13 regions flanking the *kelch13* gene shows that Wewak parasites do not cluster in a single group, and the C580Y mutants form a group on their own; no New Guinea parasites cluster with the Cambodian group, confirming that *kelch13* haplotypes on the island have independent origin. (C) IBD in chromosome 10 regions flanking the *atg18* gene shows that C580Y mutants cluster with the Timika samples, as is also the case in the chromosome 5 regions flanking the *pnp* gene (D). In B-D, flanking regions are 75 kbp regions on either flank.

where there is IBD with $\geq$ 10% of the Timika population (30% of all variants); outside of these regions, 77% of SNPs exhibit Wewak-like alleles.

Although some stretches of the C580Y genomes share IBD with both Timika and Wewak parasites, possibly indicating earlier gene flow between these populations, at several loci the IBD haplotypes are only observed in Indonesian parasites- notably in chromosomes 3, 5, 10 and 13. The sizes of these haplotypes (generally > 100 kbp) suggests recent acquisition from a donor population- putatively, one closely related to those in Indonesian Papua. The high level of IDB in these regions indicates that a large proportion of Timika parasites possess the same long haplotype, which is characteristic of genomic regions under evolutionary selection. [24]

We applied IBD analysis to other Wewak parasites, also collected in 2017, which lacked *kelch13* mutations but had high proportions of Indonesian ancestry (as determined by fastSTRUCTURE). We found that the two samples with the highest Indonesian ancestry proportions (42% and 40%) show a near-identical pattern of high-IBD regions to that of the C580Y mutants; a third parasite with a lower degree of Indonesian ancestry (29%) also harboured several corresponding high-IBD regions (S6 Fig). It therefore appears that parasites carrying very similar sets of acquired genomic regions, both with and without the C580Y mutation, circulated simultaneously in Wewak.

## Genetic variants association with *kelch13* C580Y

Because of the length of the shared IBD segments, it is not possible to pinpoint exactly what variants may have driven their acquisition. However, it is possible to narrow down the list of

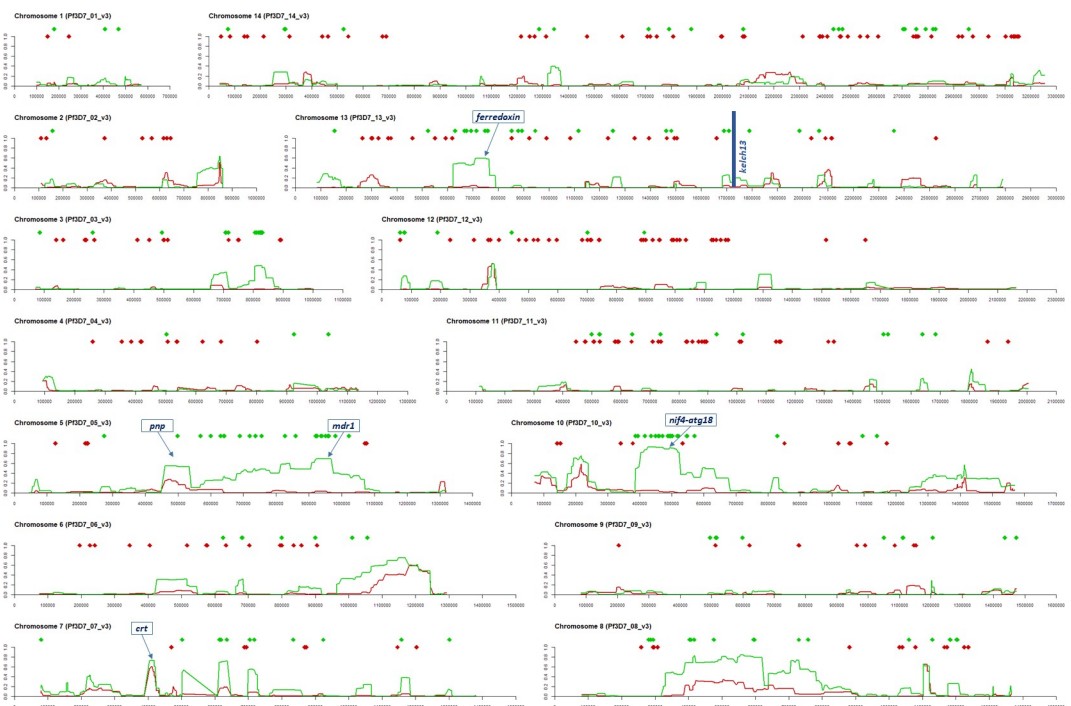

**Fig 5. Structure of genetic admixture in Wewak C580Y mutants.** Fourteen plots are shown, one per chromosome in the *P. falciparum* nuclear genome; the x-axis of each plot represents nucleotide positions along the chromosome. At the top of each plot, two rows of diamond markers indicate the position of SNPs that are highly differentiated between the Wewak and Timika populations ($F_{ST} \geq 0.3$); the marker colour indicates whether the allele present in the Wewak C580Y mutants matches the most common one in Timika (green diamonds) or Wewak (red diamonds). The line charts below the markers show the proportion of sample pairs in the Timika (green lines) and Wewak (red) population that are IDB with the Wewak C580Y parasites at each chromosome position.

potential candidates, based on the expectation that the driving mutations would cause amino acid changes, and would be common in Indonesian Papua and rare in PNG. Therefore, we extracted those nonsynonymous variants that are highly differentiated SNPs between the Wewak and Timika populations ($F_{ST} \geq 0.3$), and at which the C580Y mutants carry the Timika-like allele (n = 144, 0.6% of all nonsynonymous variants), ordering them by decreasing $F_{ST}$ (S6 Table). The resulting list of variants may be used as a shortlist for further functional investigations; it may be of particular interest that several of the most differentiated variants, particularly those in regions of high IBD with Timika parasites, have been previously implicated in drug response, or are located in genes associated with drug resistance.

The most differentiated SNP was *ferredoxin* (PF3D7_1318100) D193Y on chromosome 13, previously identified as a component of the genetic background associated with artemisinin resistance in the GMS [21]. In the large IBD region on chromosome 10 we found three noteworthy hits: the T38I mutation in the autophagy-related protein 18 gene (*atg18*, PF3D7_1012900), associated with artemisinin response and widespread in the GMS; and two SNPs (N280S and N659S) in the NLI interacting factor-like phosphatase (*nif4*, PF3D7_1012700), implicated in conjunction with *atg18* [25] and identified as associated with artemisinin response [21]. On chromosome 5, another extensive IBD region contains mutations Y184F and N1042D in the *mdr1* gene (PF3D7_0523000), which has been implicated in resistance to multiple antimalarials [26,27], including artemisinin; the Y184F variant is common in the GMS, where it has been under selection [28]. Also on chromosome 5, we found the Q225H mutations in the purine nucleoside phosphorylase gene (*pnp*, PF3D7_0513300), which

circulates in the GMS and associated with drug resistance [29]. Some highly differentiated variants are located in genes associated in drug resistance, although the mutations themselves were not directly implicated. We found K753Q in the *amino acid transporter* PF3D7_0515500, previously implicated in artemether response [30]; G248R in gene PF3D7_1450800, associated to artemisinin resistance in a large-scale study [21]; two amino acid changes in an ABC transporter (PF3D7_0319700) shown experimentally to be a drug target [31]; and the G233R mutation in *plasmepsin III* (*pmIII/hap*, PF3D7_1408100) which is involved in resistance to piperaquine [32]. IBD networks restricted to flanking regions of specific drug resistance genes such as *atg18* and *pnp* confirmed that Wewak mutants clearly cluster with the Timika population (Fig 4C and 4D).

## Discussion

Over several decades, the spread of drug resistance from the GMS has rendered multiple drugs ineffective in Africa, at the cost of hundreds of thousands of lives. Therefore, the current rise of artemisinin resistance in southeast Asia is an urgent concern, since fast-acting artemisinin combination therapies are the preferred frontline treatments in nearly all endemic countries. The GMS is not the only region with the conditions for the development of drug resistance: resistance to multiple drugs, such as chloroquine [33] and sulfadoxine [16] emerged independently in New Guinea, making the island a likely source of new resistant strains.

In this study, we identified three infections from parasites carrying the *kelch13* C580Y mutation, the most widespread artemisinin resistant allele. C580Y has rapidly overtaken other *kelch13* variants, becoming dominant in large parts of the GMS [12], and was confirmed by a transfection study to confer resistance *in vitro* [34]. The discovery of three C580Y mutants in Wewak, almost identical at genomic level, raised multiple important questions. First, are these parasites actually resistant to artemisinin? Second, were these mutants imported into PNG as a result of the recent aggressive spread of C580Y in the GMS, or are they parasites native to New Guinea that have acquired the mutation either through independent emergence, or through recombination with Asian mutants? Third, do these mutants represent a confined local phenomenon, or are they representatives of a spreading population across the island, perhaps across national borders? Fourth, if the C580Y mutations were not imported, did they result from an isolated "chance event", or from a gradual evolutionary process that selected a genetic background to boost fitness? The results presented here provide convincing evidence to answer all these questions.

Although clinical parasite clearance rates were not measured in this study, *in vitro* tests of one of the Wewak mutants showed it to be resistant to artemisinin in their ring stage, consistent with C580Y mutants elsewhere. Since no failures occurred in patients infected with these parasites after artemether-lumefantrine treatment, there is no reason to suspect that this ACT is no longer efficacious, and the clinical significance of the C580Y allele in New Guinea will be clarified by *in vivo* efficacy studies. However, parasite populations stand to benefit from decreased sensitivity even without treatment failures: slower clearance increases drug exposure, boosting the parasites' chances to develop further resistance. Recent research in the GMS showed that continued drug exposure leads to accumulation of mutations that confer higher levels of resistance, facilitating the spread of multidrug resistant lineages [12].

We found no evidence that the Wewak C580Y mutations have been imported from the GMS. At whole-genome level, the mutants appear more similar to New Guinean parasites than to Cambodian ones, and analyses of flanking haplotypes and IBD showed no evidence of acquisition from imported parasites. Overall, there is a strong case for an independent emergence of the C580Y mutation on the island of New Guinea, although our data is insufficient to pinpoint a precise geographic origin.

At genome-wide level, the Wewak C580Y mutants are substantially different from the most other PNG samples. Their genetic background is mostly "PNG-like", in that it carries alleles commonly found in Wewak, in relatively short IDB fragments suggesting long-term interbreeding. However, a substantial proportion of C580Y mutants' genome consists of long haplotypes shared with many Timika parasites, suggesting these segments were acquired from a common source population strongly connected to the Papua Indonesia parasites. Furthermore, the length and frequency of the shared haplotypes suggests that they contain regions under selection in that source population.[24] This may be consistent with a recent report that a distinct parasite subpopulation has emerged in Timika and rapidly risen in frequency to 100% of infections in 2016–2017, potentially indicating a selective sweep, [35] even though the frontline ACT is still reported to be efficacious.[36]

Although the available data does not allow precise identification of the genetic variants that drove the segments' acquisition, we catalogued alleles located in the IBD fragments that were shared with the Indonesian Papua parasites and absent from other PNG samples. In this shortlist of potential drivers, we found several variants previously associated with drug resistance (and artemisinin resistance in particular), and therefore prime candidates for further analyses. In one of these shared segments we found the *ferredoxin* D193Y mutation, identical to the artemisinin resistance-linked variant circulating in the GMS [21] and otherwise absent from PNG. The drug resistant allele *mdr1* Y184F, known to be under selection in the GMS [28], is also present in a large IBD segment, with another less common *mdr1* mutation, N1042D. *mdr1* is a transporter contributing to the efflux of toxic substances from the parasite's food vacuole, and its polymorphisms have been associated with resistance to multiple drugs. Another IBD segment on chromosome 10 harbours the *atg18* T38I variant, associated with decreased sensitivities to dihydroartemisinin and artemether (the artemisinin derivative used in PNG) in the China-Myanmar border area [25], and two non-synonymous changes in the nearby *nif4* gene; other mutations in *nif4* were previously associated with artemisinin sensitivity in two independent genome-wide association studies in Southeast Asia [21,25]. We also observed a mutation in a nucleoside phosphorylase (*pnp* Q225H), recently shown to affect response to quinine and mefloquine [29], two drugs related to lumefantrine, the partner drug used in PNG.

Taken together, the above results frame the Wewak C580Y mutants in a complex evolutionary context. This *kelch13* mutation has emerged independently in New Guinea, and was found not on an ordinary Wewak genetic background, but on one that has acquired a patchwork of sizeable genomic segments from another population. These acquisitions have introduced genomic regions that were likely to be under selection in Papua Indonesia, and contain alleles new to PNG, which have been associated to drug resistance in other parts of the world. This strongly suggests that these mutants are the product of a process driven by recombination events and selection under artemisinin drug pressure, and that the *kelch13* C580Y mutation is a component of a complex constellation of genetic changes, rather than a standalone mutation. The presence of parasites carrying a similar compendium of acquired genomic segments, but no *kelch13* C580Y mutation, supports the hypothesis that the latter is a very recent acquisition on a circulating genetic background.

There is evidence that an analogous evolutionary process has taken place in the GMS, [21] where *kelch13* mutations emerged in parasites that had acquired a complex set of genetic changes. Several of these changes were present before ACT treatment failures occurred, and it is possible that a similar process is ongoing in New Guinea. It is likely that these additional mutations play a compensatory role: although the C580Y mutation alone can confer an RSA phenotype shift, [34] it is also likely to introduce a fitness cost deriving from reduced growth at ring stage, [37] which could prevent mutant parasites from thriving in the field. Further studies will be needed to establish whether there is a common pattern to these mutations,

which could be used to detect the emergence of resistant strains from genomic surveillance data before *kelch13* mutations become established and clinical failures start to occur. The findings presented are of great concern for public health, both locally and globally. They provide evidence of artemisinin-resistant parasites within New Guinea that have acquired a complex genetic background providing a survival advantage under artemisinin drug pressure, which may have intensified recently due to decreasing *P. falciparum* transmission as a result of control interventions [38].

Such a complex genetic makeup is likely to have occurred through a gradual process, rather than in a single chance event, and it is therefore reasonable to suppose that the individuals sampled in Wewak are part of a larger evolving population. Given that no evidence of *kelch13* mutants have been found by other studies on the island, it is likely that this population has emerged recently. Genetic data from the island of New Guinea is currently insufficient to demarcate the extent of the spread of this population, which may be present on either side of the Indonesia-PNG border, or both. These parasites are a potential danger to the efficacy of ACTs in New Guinea and could constitute a threat if they are not contained. Public health authorities in PNG and collaborating research institutions are currently ramping up genetic monitoring of malaria parasites at sentinel sites, and we hope this will provide much needed detailed data to map artemisinin resistant parasites and help develop containment strategies.

## Materials and methods

### Ethics statement

Ethical approvals were obtained from the Medical Research Ethical Committee of Juntendo University, Tokyo, Japan (No. 13–016) and the Medical Research Advisory Committee of PNG National Department of Health (No. 14.22, 16.41). Patients were enrolled after obtaining written informed consent from the patients or guardians.

### Clinical blood sample collection

This study was performed as part of an ex-vivo antimalarial drugs susceptibility study, conducted in January–February 2016 and in January–February 2017 at two clinics (Town and Wirui Urban) located in Wewak, East Sepik, PNG (S1 Text) [39]. All patients >1 year old showing malaria symptoms (axillary temperature $\geq$37.5C˚, fever during previous 24 h, headache, abdominal pain, nausea and/or diarrhea) were screened by Rapid Diagnosis Test (RDT) (CareStart Malaria HRP2/pLDH COMBO Test kit, Access Bio, USA). When a *P. falciparum*-positive result was obtained, patients were enrolled after obtaining written informed consent from the patients or guardians.

Blood samples were obtained by finger prick (<2 years, 100–500 μL) or peripheral venipuncture ($\geq$2 years, 1 mL), collected into EDTA-containing tubes and immediately transferred to the Wewak General Hospital laboratories. To estimate parasitemia, thick and thin blood smears were prepared and stained with 2% Giemsa for 30 min. For molecular analysis, blood samples were transferred onto chromatography filter paper (ET31CHR; Whatman Limited, Kent, UK), dried and stored at -20˚C in individual plastic bags.

This study enrolled 257 patients: 129 at Wirui Urban clinic and 128 at Town clinic (Table 1). Most patient characteristics were similar at the two clinics, except for the frequency of pre-treated patients, significantly higher at Town clinic (11.7% vs 2.3%, *P* = 0.0032, Fisher exact test). Eighteen patients reported ingesting antimalarials in the previous two weeks and were excluded from further analysis; artemether (n = 7) and chloroquine (n = 7) were the most used forms of self-medication. No severe case was enrolled.

## Genotyping and whole-genome sequencing

Genomic DNA was extracted from a quarter of blood spot (25 μL) using the QIAamp DNA blood Mini Kit (QIAGEN, Hilden, Germany) with a modified procedure [40]. The *kelch13* gene propeller domain sequence was determined by capillary sequencing of PCR products, as previously described [41]. Extracted DNA underwent selective whole genome amplification [42] prior to whole-genome sequencing, to enrich parasite DNA. Sequence data were generated with Illumina short-read technology and deposited at the European Nucleotide Archives (S7 Table). Read counts at 1,043,334 biallelic SNPs were called with the MalariaGEN *Plasmodium falciparum* Community Project V6.0 analysis pipeline [43,44]. Genotypes were called only with a coverage of five or more reads, and alleles were disregarded if supported by fewer than 2 reads, or 5% of reads when coverage was >50. To minimize errors and biases, we excluded samples with insufficient coverage at more than 25% of the SNPs (with the exception of one Wewak C580Y mutant sample), and removed all SNPs that either were invariant after those samples were removed, or had insufficient coverage in more than 25% of the remaining samples, leaving 35,664 SNPs used in our analysis.

For comparisons, we used samples included in the MalariaGEN *P. falciparum* Community Project 6.2 release (https://www.malariagen.net/projects/p-falciparum-community-project) sampled from Madang (contributed by IM), Maprik and Alotau (contributed by AB) in PNG; Indonesia samples from Timika (contributed by RN); and Cambodian samples from Pailin and Pursat provinces, contributed to the Pf3K reference dataset (https://www.malariagen.net/projects/Pf3k) by the Tracking Artemisinin Resistance Collaboration (TRAC) [3].

To minimize errors in haplotype reconstruction due to mixed infections, we filtered samples by their $F_{WS}$ index. By removing samples with $F_{WS}<0.95$, we obtained a final set of 389 essentially monoclonal samples. To estimate $F_{WS}$, allele frequencies in each population, and the differentiation measure $F_{ST}$, we applied previously published methods [21,44].

## Ex-vivo ring-stage survival assay (RSA)

Ex-vivo RSA was performed as previously described [20,41] on samples with ≥0.1% parasitemia. Parasitemia were adjusted to 1% by adding uninfected type O erythrocytes. Parasite culture mixture (100 μL per well) was dispensed with 700 nmol/L dihydroartemisinin (Tokyo Chemical Industry Co. Ltd., Tokyo, Japan). After 6 h exposure, pellets were washed to remove the drug, and incubated for 66 h. 0.1% dimethyl sulfoxide was used as control. Densities of viable parasites were determined by two investigators (MS and TM) by counting 10,000 erythrocytes, and survival rates calculated as ratios of parasites in exposed and non-exposed cultures [20]. Laboratory adapted artemisinin-susceptible (3D7) and resistant clones (MRA-1236 and MRA-1240, contributed by Didier Ménard of Institut Pasteur du Cambodge), were provided by MR4 through BEI Resources (NIAID, NIH), and used as comparators for ex-vivo RSA [20].

## Determinations of *kelch13* flanking haplotypes

**Using microsatellites and selected SNPs.**   We genotyped *kelch13* flanking microsatellite loci and single-nucleotide polymorphisms (SNPs) located at -33.9 kb, -8.1 kb, -5.02 kb, -1.91 kb, 4.1 kb, 9.33 kb, 15.82 kb, and 43.85 kb from the *kelch13* gene (negative and positive values denote upstream and downstream distance from *kelch13*, respectively) by direct sequence as previously described [14,23]. We also genotyped four KEL1/PLA1 marker SNPs located at -137.5 kb, -14.06 kb, 6.97 kb, and 24.91 kb [13]. At positions -137.5 kb and 24.91 kb, SNPs were determined by direct sequencing of PCR products. At position 6.97 kb, SNP-specific amplification was performed with HiDi DNA polymerase (myPOLS Biotec, Konstanz, Germany). At position -14.06 kb, SNP was detected by combination of Derived Cleaved Amplified

Polymorphic Sequences (dCAPS) assay and PstI digestion. Primers used for these analyses are shown in S8 Table.

**Using whole-genome sequencing data.** We constructed flanking haplotypes by concatenating WGS genotypes at SNPs with minor allele frequency (MAF) $\geq 0.01$ in a region of 300kbp centred at *kelch13*.

**Comparison and ranking of *kelch13* flanking haplotypes.** To assess flanking haplotypes, we assigned to each sample a score based on the extent of haplotype sequence identity with the consensus haplotype of the Wewak *kelch13* mutants. Foreach flank, we started from the position nearest to the *kelch13* gene, and moving away from *kelch13*, counted the number of consecutive SNPs (ignoring missing/heterozygous positions) that carried an allele identical to that of C580Y mutants, until a mismatch was found. The final score for the sample was obtained by adding the two flank scores.

**Genotyping of background mutations associated with artemisinin resistance.** Genotyping of background alleles associated with artemisinin resistance (D193Y in *fd*, T484I in *mdr2*, V127M in *arps10*, I356T in *crt*, V1157L in *nif4* and C1484F in *pibp*) [21] were performed by multiplex PCR and probe qPCR as previously described [45]. Copy number variants of *plasmepsin2/3* and *mdr1* genes were determined using published procedures [46]. Four artemisinin-resistant clones (MRA-1236, MRA-1238, MRA-1240 and MRA-1241) and three Cambodian isolates [16] were used as comparators.

## Analysis of ancestry and relatedness

Analyses of population structure were performed using freely available tools and custom software programs written in Java and R. Identity by descent (IBD) analysis was performed on the genotypes obtained from whole-genome sequencing using hmmIBD [47] with default parameters. Further estimates of IBD, producing IBD networks, were generated by isoRelate [48]. We constructed an *NxN* pairwise genetic distance matrix (*N* is the number of samples). estimating genome-wide genetic distance using a previously published procedure [8]. In addition, we constructed an *NxN* pairwise IBD distance matrix, estimating the pairwise IBD distance as $d_{IBD} = 1 - p_{IBD}$, where $p_{IBD}$ is the fraction of the genome predicted to be identical by descent. PCoA analyses were performed using the R stats package (https://www.r-project.org/), while neighbour-joining trees were produced by the ape package [49]. Admixture analyses were performed using the fastSTRUCTURE software in Bioconda [50,51]. We ran fastSTRUCTURE with K = 2 to 9 putative ancestral populations, and ran chooseK to determine the value of K that best explains the population structure (K = 6).

## Supporting information

**S1 Fig. Approximate locations of residence of patients infected with *kelch13* C580Y mutant parasites.** This map of Wewak town shows the place of abode of the three patients whose parasites carried the *kelch13* C580Y allele (red markers), the distance between these locations (blue lines), and the location of the two clinics where the study was carried out (green markers) (TIF)

**S2 Fig. RSA survival rates of Wewak parasites.** The plot compares the RSA survival rate (see Methods) of one of the Wewak *kelch13* C580Y mutants (left, red marker) against those for wild-type parasites from the same area (right, black markers). RSA survival rates could not be determined for the remaining two Wewak *kelch13* C580Y mutants. Artemisinin-susceptible laboratory clone 3D7 showed no parasite at 700 nmol/L. MRA-1236 and MRA-1240 (artemisinin-resistant laboratory **clones) showed survival rates 14.3% and 27.0%, respectively.** (TIF)

**S3 Fig. Principal coordinates analysis (PCoA) based on genome-wide pairwise genetic distances.** This figure shows a plot of the first two components, for all analyzed samples. The first component (PC1), which explains most of the dataset variance, separates samples from Cambodia from those in New Guinea. The second component (PC2) separates parasites from Papua Indonesia from those found in PNG. We note that there is some overlap between these two groups, and the Wewak C580Y parasites (yellow) appear at an intermediate point between PNG and Indonesian parasites.
(TIF)

**S4 Fig. Population structure in the analyzed sample set.** The plots show admixture levels estimated by the fastSTRUCTURE software, based on the hypothesis of *K* ancestral populations (*K* = 2 to 9, shown on the left-hand side of the plot). Each population is represented by a different colour (*K* arbitrary colours for each plot); each sample is shown as a vertical bar, coloured according to the proportion of ancestry from each population. Samples are grouped by sampling location, as shown by labels at the bottom; Wewak C580Y mutants are shown as a separate group. From an analysis of the underlying data, the fastSTRUCTURE chooseK tool reported that *K* = 6 is that number of populations that best explains the population structure.
(TIF)

**S5 Fig. Comparison of haplotypes in a ±150 kbp region on chromosome 13, flanking the *kelch13* gene.** Each row represents the haplotype of a sample, the top row showing the consensus haplotype for the C580Y samples in Wewak. Each column represents a variant position; the position of the *kelch13* C580Y mutations (1725259) is indicated by a box outline. Cells colours show the allele call at each position in the sample. Deep colour hues indicate a matching haplotype portion (i.e. consecutive positions within the flanking haplotypes that match the consensus haplotype), while lighter colours indicate positions after a haplotype mismatch. Blue cells symbolize the reference allele, orange the alternative allele, and gray denote a mixed allele call or insufficient coverage. Samples are grouped by country of provenance: Cambodia (panel A), Papua New Guinea (B), Indonesia (C), and sorted by decreasing matching score (the sum of length of the matching haplotype portions in the two flanks). The column on the left shows the sample identifier, while the columns on the right show the matching score and the *kelch13* allele carried by the sample.
(TIF)

**S6 Fig. Correspondence between IBD regions in Wewak C580Y mutant and those in Wewak *kelch13* wild-type parasites with high proportion of Timika ancestry.** These plots show, across all nuclear chromosomes, the proportion of IBD pairs between Timika and the Wewak C580Y mutants (red, top), and three Wewak samples that showed a high proportion of common ancestry with Timika: one with 42% Timika ancestry (blue), one with 40% (magenta) and one with 29% (green). The top panel shows vertical green bars marking highly differentiated positions where the C580Y mutant carry a Timika-like allele. Green diamond markers show the location of some notable drug resistance-related alleles identified in our analysis. Vertical dotted lines act as visual guides to show correspondences.
(TIF)

**S1 Table. Genotypes at loci of resistance for ACT component drugs.** The counts of samples carrying wild-type and mutant alleles at three drug resistance-associated loci are shown by collection season (2016 and 2017).
(XLSX)

**S2 Table. Characteristics of the three Wewak *kelch13* C580Y mutant parasites.** The table shows date of collection, patient characteristics, clinical symptoms, in vitro results and

genotypes at drug resistance-associated loci for the three Wewak samples carrying the *kelch13* C580Y mutation.
(XLSX)

**S3 Table. Ring-stage survival assay results.** The counts of samples which produces in vitro RSA results, and of those that exhibited higher survival rates, are shown by collection season (2016 and 2017).
(XLSX)

**S4 Table. Amino acid alleles at loci in the artemisinin resistance genetic background.** The table includes positions previously associated with *kelch13* mutations in the GMS (see main text). Samples prefixed with "MRA" originated from the GMS, while those prefixed "PNG" were sampled in Wewak. Coloured background denoted alleles different from the reference (wild type). Light blue background reflects a mutant allele different from that previously reported in the GMS.
(XLSX)

**S5 Table. Haplotypes in the *kelch13* flanking region on chromosome 13, based on genotyping 18 microsatellites and SNPs.** The locations of marker sites are described in terms of their distance from the *kelch13* gene (negative and positive values denote upstream and downstream distances respectively). Colour backgrounds denote an uninterrupted sequence of alleles identical to one of the Wewak C580Y haplotypes, in either direction as one traverses from the *kelch13* gene. Reference isolates prefixed with "CAM" and "MRA" originated from the GMS. The Wewak wild type haplotypes are sorted by the total length of the identical genotype sequences.
(XLSX)

**S6 Table. Highly differentiated SNPs where Wewak C580Y mutants and Papua Indonesia parasites carry the same allele.** Each row shows a non-synonymous SNP that is highly differentiated between the Wewak (PNG) and Timika (Indonesia) populations ($F_{ST} \geq 0.3$) at which the Wewak C580Y mutants carry the allele that is most prevalent in Timika. The columns show: the chromosome number and nucleotide position of the SNP; the name and ID of the gene containing the SNP (genes associated with drug resistance are shown on a coloured background); the mutation name, showing amino acid change and position; the frequency of the mutant allele, in Wewak and in Timika (background colour intensity is proportional to the value); the $F_{ST}$ between the two populations; the proportion of sample pairs that are in IBD with Wewak C580 samples at that position, in Wewak and Timika (background colour intensity is proportional to the value); and the Pubmed PMID reference of literature that implicates the gene in drug resistance, if applicable.
(XLSX)

**S7 Table. European Nucleotide Archive (ENA) identifiers for samples used in the present study.** For each sample in PNG and the GMS, we detail: the MalariaGEN sample identifier; the location and country where the sample was collected; the identifier of the sample at ENA (https://www.ebi.ac.uk/ena) where the sequencing reads for the sample have been deposited; and the label used in this paper for Wewak samples that carry the C580Y mutation in *kelch13*.
(XLSX)

**S8 Table. Primers used for determination of *kelch13*-flanking SNPs.** Sequences and role are shown for primers used to genotype four SNPs in *kelch13* flanking regions.
(XLSX)

**S1 Text. Description of sampling location.** A description of the climatic, entomological and epidemiological characteristics of the Wewak location, with a description of recent malaria control interventions.
(DOCX)

## Acknowledgments

We thank all patients who contributed samples, and their guardians; Steven Tiwara, Alpha Ao, Charlie Amai, Alphonse Coll, Douglas Tambi, Gethrude Kitipa, Julianne Gumbat, Raphael Kariwa, and Shoki Yatsushiro for their kind cooperation in the field; Takahiro Tsukahara for technical advice; the Laboratory of Molecular and Biochemical Research, Research Support Center, Juntendo University Graduate School of Medicine, for technical assistance; the staff of the PNG Institute of Medical Research and Abby Harrison for their contributions to sample collection. Genome sequencing was performed by the Wellcome Sanger Institute (WSI), and sequencing data was processed by the MalariaGEN Resource Centre; this study used data from the MalariaGEN Pf3k Project and Plasmodium falciparum Community Project. We thank the staff of the WSI Sample Logistics, Sequencing, and Informatics facilities for their contribution; Mihir Kekre and Katie Love for their support in the sample processing pipeline; Victoria Cornelius and Kim Johnson for coordinating the MalariaGEN Resource Centre; the TRAC investigators who provided the Cambodian isolates used in this work, including Elizabeth Ashley, Rick M Fairhurst, Chanaki Amaratunga, Arjen Dondorp and Nicholas J White. PR is a staff member of the World Health Organization; PR alone is responsible for views expressed in this publication and they do not necessarily represent the decisions, policy or views of the World Health Organization.

## Author Contributions

**Conceptualization:** Olivo Miotto, Toshihiro Mita.

**Data curation:** Olivo Miotto, Richard D. Pearson, Roberto Amato, Toshihiro Mita.

**Formal analysis:** Olivo Miotto, Somya Mehra, Alyssa E. Barry, Jun Ohashi.

**Funding acquisition:** Dominic P. Kwiatkowski, Toshihiro Mita.

**Investigation:** Olivo Miotto, Makoto Sekihara, Shin-Ichiro Tachibana, Masato Yamauchi, Richard D. Pearson, Roberto Amato, Sonia Gonçalves, Somya Mehra, Rintis Noviyanti, Jutta Marfurt, Sarah Auburn, Ric N. Price, Ivo Mueller, Mie Ikeda, Toshiyuki Mori, Makoto Hirai, Manuel W. Hetzel, Alyssa E. Barry, Pascal Ringwald, Jun Ohashi, Toshihiro Mita.

**Methodology:** Alyssa E. Barry, Jun Ohashi.

**Project administration:** Olivo Miotto, Toshihiro Mita.

**Resources:** Makoto Sekihara, Shin-Ichiro Tachibana, Masato Yamauchi, Richard D. Pearson, Roberto Amato, Sonia Gonçalves, Rintis Noviyanti, Jutta Marfurt, Sarah Auburn, Ric N. Price, Ivo Mueller, Livingstone Tavul, Manuel W. Hetzel, Moses Laman, Alyssa E. Barry, Jun Ohashi, Dominic P. Kwiatkowski, Toshihiro Mita.

**Software:** Olivo Miotto, Richard D. Pearson.

**Supervision:** Rintis Noviyanti, Livingstone Tavul, Moses Laman, Alyssa E. Barry, Jun Ohashi, Francis Hombhanje, Dominic P. Kwiatkowski.

**Validation:** Makoto Sekihara, Somya Mehra, Toshihiro Mita.

**Visualization:** Olivo Miotto, Shin-Ichiro Tachibana, Somya Mehra, Toshihiro Mita.

**Writing – original draft:** Olivo Miotto, Makoto Sekihara, Toshihiro Mita.

**Writing – review & editing:** Olivo Miotto, Makoto Sekihara, Alyssa E. Barry, Jun Ohashi, Toshihiro Mita.

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
