## [Decision Letter · Decision Letter 0]

12 Aug 2020

Dear Dr. Miotto,

Thank you very much for submitting your manuscript "Emergence of artemisinin-resistance Plasmodium falciparum with kelch13 C580Y mutations on the island of New Guinea" for consideration at PLOS Pathogens. As with all papers reviewed by the journal, your manuscript was reviewed by members of the editorial board and by several independent reviewers. The reviewers appreciated the attention to an important topic. Based on the reviews, we are likely to accept this manuscript for publication, providing that you modify the manuscript according to the review recommendations.

We would ask you to please provide the clarifications requested by reviewer 1 and also to consider carefully the comments by reviewer 2, who wonders how your data fit with our growing knowledge of artemisinin resistance mechanisms, with any clinical and *in vitro* phenotypes observed in PNG and the particular drug combinations used in the area. We agree that a more explicit discussion of these aspects, as well as of the concrete implications of your findings for surveillance, would greatly add to the value of your paper and increase its interest to a broader audience. 

Sincerely,

Oliver Billker

Associate Editor

PLOS Pathogens

Kirk Deitsch

Section Editor

PLOS Pathogens

Kasturi Haldar

Editor-in-Chief

PLOS Pathogens

orcid.org/0000-0001-5065-158X

Michael Malim

Editor-in-Chief

PLOS Pathogens

orcid.org/0000-0002-7699-2064

Reviewer Comments (if any, and for reference):

Reviewer's Responses to Questions

**Part I - Summary**

Reviewer #1: I enjoyed this manuscript. It presents good evidence for an independent origin of artemisin resistance in PNG, showing it is an independent haplotype.

Reviewer #2: This is important and exciting work from arguably the world leaders in evolutionary genomics and molecular epidemiology of malaria parasites and drug resistance. The work is impressive both in terms of the study/data/analysis and in terms of the thoughtful presentation. However, while what the authors DO describe is compelling, there are important things they are not describing and putting in broader context. They stop short of using their study to address important unresolved questions about Art-R. Stopping short of clarity would be a missed opportunity to make the most of this work to provide a template that can serve to guide the field in the important and challenging area of artemisinin drug resistance (Art-R) evolution, particularly as relates to surveillance and to have a coherent scheme for decisions and policymaking. To this end, what does this work indicate for researchers on the front lines who are monitoring for emerging Art-R? The authors are having it both ways. What is/are the signature that matters? Is K13 (kelch) all we need to know? Or is that the final step to Art-R and what matters is a source of ‘background’ with specific features? And what about the phenotype limitations of this study? Without the single C580Y parasite for which they have RSA data (indeed, it is their most resistant RSA of all successfully measured), the message of this paper would be largely muted. And they give nearly no consideration to the 5 RSA resistant parasites that carry a kelch wild type (WT). Is the phenotype uninteresting in this case? Are these 2/3 of the way to resistant (but still lacking the kelch mutation). There is an opportunity for this paper to be both excellent (with minor revisions) and pivotal (with some deeper consideration of its own limitations and what would be needed to get definitive and crucial answers).

**Part II – Major Issues: Key Experiments Required for Acceptance**

Reviewer #1: (No Response)

Reviewer #2: (No Response)

**Part III – Minor Issues: Editorial and Data Presentation Modifications**

Reviewer #1: Some part of the narrative seems a bit like story telling. The high Fst SNPS that were introgressed being associated with resistance. What is the Fst distribution between Indonesia and PNG? What is the extent of the evidence that the high Fst SNPs are likely to have been selected in Indonesia, could they be introgressed from elsewhere? Are there other possible sources of introgression.

I think the ambiguity about evolutionary scenario is somewhat inevitable since really it hinges on the demographic history in Indonesia and elsewhere which is not the topic of the present manuscript. It should be made as clear as possible which part of this argument is well supported by data and which is more indirect inference. Most important, what is the evidence that this association with resistance, which is interesting if true whatever the demographic explanation is really statistically surprising?

Looking at figure Table S5, it seems the evidence that its an independent haplotype seems to come only from the right hand flank of the gene? Am I correct in this? If so is it mentioned and discussed in the text? Could it possibly be a recombinant?

This manuscript suggests that resistance evolves by multiple steps but an analysis of patterns over time in multiple locations would be really necessary to get a firm handle on this. As far as possible, it would be nice if this discussion could be walled off as more speculative. Are there specific predictions about the specific patterns associated with resistance loci that the authors would like to make that could be investigated in a global way?

Reviewer #2: Broadly, there are unaddressed items that will leave an uninitiated reader confused. For example:

The growing body of literature implies that a mutation in K13, specifically the most successful C580Y mutation, is the Art-R determining mutation. Recent work by Birnbaum et al reinforced this with their compelling (and surprising) functional basis for Art-R. The emerging story would thus indicate that Art-R evolution is simple, sweep signatures should be generally hard, clinical phenotypes unambiguous and that K13 mutations is the only relevant surveillance. This manuscript nicely illustrates that it is more nuanced, but stops short of taking on any of these points directly. Because widespread Art-R would be a devastating blow to malaria control, researchers around the world are groping for how best to identify emerging resistance and rapidly respond with smart adjustments to local policies.

Why is there no treatment failure or evidence of delayed clearance on the island? Is this a feature of the local malaria? The specific activities in the clinics? The phenotypes per se? e.g. Are these ex vivo RSA data reliable? Is RSA a reliable surrogate for clearance times? And are both underpinning drug failure? Are there unique aspects to malaria, clinical access, reporting that could influence this interpretation (e.g. of hidden emerging resistance)?

Are the findings of Birnbaum et al relevant to understanding/interpreting these data? Not relevant at all? Why?

Timiki is crucial for reconstructing the current epidemiological situation in Wewak C580Y mutants. The authors give very little information about what is special about these parasites, e.g. the drug (combination) of choice in that region, duration (and sequence), intensity of selection, clinical situation, the possible source of introduction of this background.

The authors describe these genome segments as “recently acquired and possibly under selection”; explain this further (particularly the inference and role of selection, before and after introgression). I appreciate the authors hesitate to speak beyond their data, but there is room for more clarity about why this matters.

Besides C580Y, it seems there are no other K13 mutations to mention?

It would be useful to more directly compare the findings here with a similar recent report from Guyana; what do these works say about expectations with respect to Art-R appearing in Africa? What information will be needed in future studies to get definitive actionable results. What is the recommendation based on this study for ongoing surveillance, in PNG and around the world?

Other intriguing statements need clarification: “gradually acquired a complex ensemble of variants” (in what order? Can C580Y exist absent the background? Is the RSA phenotype shift caused by only C580Y?). Is this what the authors mean by “not a chance event”? i.e. that there is no C580Y by itself? But otherwise, it seems there is no specific mutation diagnostic of Art-R, but perhaps there is a signature emerging of some sort?

How can this method (e.g. IBD, key background components, K13 haplotypes) be systematically considered going forward, across different geographies by different groups? If groups are content to sequence kelch, will that provide the necessary real time information to adjust to newly arriving/emerging Art-R. The authors say (Art-R) “went undetected on the island”. What is predicted? What is advised? The last sentence in the abstract implies this information will be useful to “to contain the spread”; explain a bit more how this would work.

These sentences from the Discussion are a good attempt, but are cryptic, with several opportunities to speak more specifically (comments interspersed). “We propose that one or more New Guinean lineages have accumulated a complex genetic background through recombination events and selection under artemisinin drug pressure (necessarily art pressure? Do particular partner drugs play a crucial role? All are CQR? Is that (or other selection histories) relevant?), and that the kelch13 C580Y mutation is a component of a complex constellation of genetic changes, rather than a standalone mutation generated by a chance event (what sort of component? essential, necessary? Must come last? due to fitness limitations? Is there a pre- or low-level resistance?).”… “it appears that several of these changes were present before ACT treatment failures occurred (my understanding is there is no report of treatment failure in PNG, is this referring to GMS? Does this scenario align with recent Guyana findings?), suggesting they provided improved fitness of the parasite population without major effects on clinical outcomes (are you confident that clinics were well-tuned to identify this?). In scenarios (do we have examples of this with other drugs/pathogens?) where drug resistance requires a gradual build‐up of genetic changes, clinical efficacy monitoring is only likely to detect resistance at an advanced stage of the evolutionary process (major implications here about phenotypes…be clear what is needed), when failures begin to occur. In the future, it is possible that novel analyses of evolutionary patterns, using genomic surveillance data, could allow earlier detection of drug resistance, and interventions ahead of clinical failure (yes, this is the one that could move the field forward if the authors will speak more clearly…what would this have to look like?).

Specific comments

Some additional details needed:

1. In figure 2B the fastStructure analysis shows a small subset of parasites from Wewak have as much or more Timiki ancestry as the C580Y parasites. Do any of the parasite in this subset also have the predominantly Timiki alleles at the same alleles that are identified in large sections from the C580Y parasites that are IBD with Timiki parasites? For example, in Figure 4C (the IBD plot for the region surrounding atg18) it appears that for some Wewak parasites, in addition to the C580Y parasite, that cluster with the Timiki parasites; however, this does not seem to be the case for flanking the pnp gene.

2. Do the authors think these variants have been under selection in Timiki? They give little information about ‘why Timiki’ as the source of background, e.g. in terms of drug selection history, current ACT, any clinical information.

3. Define clinical outcomes in both PNG and Indonesia Papua…has there been treatment failure? Clearance rates were not measured, true? So the authors can’t speak to the impact of these mutations on clearance rate specifically.

4. It needs to be addressed more directly (as it is highly relevant to surveillance)… if it is true that there is no treatment failure, including for the kelch mutants, there is not a strong ground based on these studies alone to focus entirely on the (assume) the role of kelch, and e.g. to discount the other RSA resistant kelch WT parasites, yet there is no mention of these parasites (or specific consideration of their Timiki IBD; are these one step toward resistant? Awaiting only a C580Y mutation?

5. In Table S3 they have survival rate greater than 0% but the RSA cutoff is 1% and they state that in line 121, why are these not consistent? Do the RSA resistant kelch WT share any specific mutations they list in figure S6 between the C580Ys? There might be value to highlighting these in the PCoA in Fig2. Are there candidate parasites that ‘bridge the gap’ between the clusters from the two populations? Do the samples H5, H13, H46 and H48 parasites (lines 172-174) share sequence similarities?

Minor comments:

Lines 165-176, 177-183, 201-206 show that the haplotype didn’t come from Cambodia; does this rule out other SEA sources?

Line 215, might be better to say “chosen” rather than “selected”.

Line 201-204/figure 4. They have 3 Wewak C580Ymutants but only show 2 points in their clusters. Enlarge the circles and squares to make clear what is grayed out for the figure key to be more clear.

For Fig 5 and lines 207-227 indicate Timika segments are selected by fitness advantage or drug pressure. How do the authors envision their Fig 5 will look in the future? Will those section of chromosome that show high IBD with Timika parasite be whittled down even further, if those mutations work better with C580Y to make drug resistant parasites? How would that graph look with those WT parasites?

 

For TableS6 Mutation Frequency, ID needs to be Timika.

Line 278 – showed that one Wewak mutant, not “mutants”, are resistant to art.

Can efficacy studies detecting delayed PC1/2 demonstrate advanced resistance? A point they could make that would help this approach is whether there is enough new cases happening in PNG to make efficacy studies happen. If there isn’t, the only way to really monitor resistance is to do in vitro RSA and sequencing?

These low transmission areas seem susceptible to Art-R generation. It would be useful to consider whether this information will be relevant for Art-R in Africa.

PLOS authors have the option to publish the peer review history of their article (what does this mean?). If published, this will include your full peer review and any attached files.

Reviewer #1: No

Reviewer #2: No
---

## [Decision Letter · Decision Letter 1]

5 Nov 2020

Dear Dr. Miotto,

We are pleased to inform you that your manuscript 'Emergence of artemisinin-resistance Plasmodium falciparum with kelch13 C580Y mutations on the island of New Guinea' has been provisionally accepted for publication in PLOS Pathogens.

Best regards,

Oliver Billker

Associate Editor

PLOS Pathogens

Kirk Deitsch

Section Editor

PLOS Pathogens

Kasturi Haldar

Editor-in-Chief

PLOS Pathogens

orcid.org/0000-0001-5065-158X

Michael Malim

Editor-in-Chief

PLOS Pathogens

orcid.org/0000-0002-7699-2064

Reviewer Comments (if any, and for reference):

Reviewer's Responses to Questions

**Part I - Summary**

Reviewer #2: The authors have done an impressive job addressing all of my concerns and providing detailed explanations for items that they argued effectively could not be supported by the data. I appreciate the detail and care of the rebuttal and am confident this revised manuscript will be an important advance (and new standard) for the field of malaria parasite drug resistance genetics.

**Part II – Major Issues: Key Experiments Required for Acceptance**

Reviewer #2: (No Response)

**Part III – Minor Issues: Editorial and Data Presentation Modifications**

Reviewer #2: (No Response)

PLOS authors have the option to publish the peer review history of their article (what does this mean?). If published, this will include your full peer review and any attached files.

Reviewer #2: No

---

## [Editor Report · Acceptance letter]

4 Dec 2020

Dear Dr. Miotto,

We are delighted to inform you that your manuscript, "Emergence of artemisinin-resistance Plasmodium falciparum with kelch13 C580Y mutations on the island of New Guinea," has been formally accepted for publication in PLOS Pathogens.

Best regards,

Kasturi Haldar

Editor-in-Chief

PLOS Pathogens

orcid.org/0000-0001-5065-158X

Michael Malim

Editor-in-Chief

PLOS Pathogens

orcid.org/0000-0002-7699-2064